# Modeling and Predicting Pulmonary Tuberculosis Incidence and Its Association with Air Pollution and Meteorological Factors Using an ARIMAX Model: An Ecological Study in Ningbo of China

**DOI:** 10.3390/ijerph19095385

**Published:** 2022-04-28

**Authors:** Yun-Peng Chen, Le-Fan Liu, Yang Che, Jing Huang, Guo-Xing Li, Guo-Xin Sang, Zhi-Qiang Xuan, Tian-Feng He

**Affiliations:** 1School of Medicine, Ningbo University, 818 Fenghua Road, Ningbo 315211, China; chenypnbu@163.com; 2Center for Health Economics, School of Economics, University of Nottingham Ningbo China, 199 Taikang East Road, Ningbo 315100, China; lefan.liu@nottingham.edu.cn; 3Institute of Tuberculosis Prevention and Control, Ningbo Municipal Center for Disease Control and Prevention, 237 Yongfeng Road, Ningbo 315010, China; 13805876046@163.com (Y.C.); sanggxnbcdc@163.com (G.-X.S.); 4Department of Occupational and Environmental Health Sciences, School of Public Health, Peking University, 38 Xueyuan Road, Beijing 100191, China; jing_huang@bjmu.edu.cn (J.H.); liguoxing@bjmu.edu.cn (G.-X.L.); 5Institute of Occupational Health and Radiation Protection, Zhejiang Provincial Center for Disease Control and Prevention, 3399 Binshen Road, Hangzhou 310051, China

**Keywords:** pulmonary tuberculosis, air pollution, meteorological factor, time series

## Abstract

The autoregressive integrated moving average with exogenous regressors (ARIMAX) modeling studies of pulmonary tuberculosis (PTB) are still rare. This study aims to explore whether incorporating air pollution and meteorological factors can improve the performance of a time series model in predicting PTB. We collected the monthly incidence of PTB, records of six air pollutants and six meteorological factors in Ningbo of China from January 2015 to December 2019. Then, we constructed the ARIMA, univariate ARIMAX, and multivariate ARIMAX models. The ARIMAX model incorporated ambient factors, while the ARIMA model did not. After prewhitening, the cross-correlation analysis showed that PTB incidence was related to air pollution and meteorological factors with a lag effect. Air pollution and meteorological factors also had a correlation. We found that the multivariate ARIMAX model incorporating both the ozone with 0-month lag and the atmospheric pressure with 11-month lag had the best performance for predicting the incidence of PTB in 2019, with the lowest fitted mean absolute percentage error (MAPE) of 2.9097% and test MAPE of 9.2643%. However, ARIMAX has limited improvement in prediction accuracy compared with the ARIMA model. Our study also suggests the role of protecting the environment and reducing pollutants in controlling PTB and other infectious diseases.

## 1. Introduction

Tuberculosis (TB) is a chronic infectious disease caused by *Mycobacterium tuberculosis*, and it often affects the lungs. WHO proposed an End TB Strategy in 2014, with the targets to reduce TB deaths by 95% and to cut incident cases by 90% between 2015 and 2035 [1]. The cumulative reduction in TB incidence from 2015 to 2020 was 11%, just over half-way to the 2020 milestone of the strategy [2]. To achieve this ambitious goal, accurate prediction of disease trends, as well as related factors, is of great importance.

The autoregressive integrated moving average (ARIMA) model, also known as the Box–Jenkins model, is a commonly used model in time series analysis. Despite its effectiveness to study time series, the ARIMA model applies only to one variable. The autoregressive integrated moving average with exogenous regressors (ARIMAX) model, however, adds other variables related to the target series as input variables to improve the prediction accuracy. Unlike the ARIMA model, the autoregressive integrated moving average with exogenous regressors (ARIMAX) model adds other variables related to the target series as input variables to improve the prediction accuracy. Several ARIMAX studies suggested that weather parameters such as temperature, rainfall, humidity, wind speed, and air pollutants may influence the occurrence of disease [3,4,5,6,7]. ARIMAX modeling studies of tuberculosis are still rare. A time series study in Jiangsu, China, showed that the prediction accuracy of the ARIMA model for PTB was improved by adding monthly PM_2.5_ with 0-month lag as an external variable [8]. Another time series study in eastern China also indicated that the predictive performance of the ARIMA model was improved after incorporating meteorological factors [9].

Several studies have proposed the biological mechanisms linking air pollution exposure and the risk of PTB. For example, PM_2.5_ exposure disrupted the synthesis and secretion of inflammatory cytokines including interferon (IFN)-γ, tumor necrosis factor (TNF)-α, and interleukin (IL)-10 and impaired key anti-mycobacterial T cell immune functions [10]. In addition, blocking the IL-10 pathway and downregulating TNF-α by CO in lung macrophages may promote the reactivation of PTB [11]. Therefore, we predicted a positive correlation between pollutant concentration and tuberculosis incidence. The possible link between PTB and meteorological factors may be attributable to the following reasons. The risk of *Mycobacterium tuberculosis* transmission appears to be the greatest during the cold winter, particularly in overcrowded and poorly ventilated settings [12]. Less humidity leads to the evaporation of droplets, reduces their size, and escalates their ability to travel further, which increases the possibility of transmission [13]. The atmosphere pressure has an indirect negative effect on the incidence of PTB. Low-level air rises under low-pressure conditions, and surface pollutants diffuse vertically into the air, resulting in increased air pollution. Therefore, we predicted a negative correlation between meteorological factors and tuberculosis incidence [14].

To our knowledge, few studies used the ARIMAX model to incorporate both the air pollution and meteorological factors to predict PTB. Thus, in the current study, we performed a time series analysis in Ningbo, China, and applied ARIMA models (ARIMA, univariate ARIMAX and multivariate ARIMAX model) to explore whether the inclusion of air pollution and meteorological factors can improve the performance of prediction modeling.

## 2. Materials and Methods

### 2.1. Study Site and Data Collection

As a city in Zhejiang province located along the eastern coast of China, Ningbo covers an area of 98,000 thousand square kilometers. It governed 10 counties and had a permanent population of 9.4 million at the end of 2020. All newly diagnosed TB cases are registered in an online Tuberculosis Management Information System (TBIMS), which is operated by the Center for Disease Control and Prevention (CDC) of China. We extracted monthly incidence of PTB from January 2015 to December 2019 as the study subjects. Population data were obtained from the Ningbo Statistical Yearbook. We used the monthly incidence from January 2015 to December 2018 as the model-construction datasets and incidence from January 2019 to December 2019 as the validation datasets.

The monthly average concentrations of the ambient air pollutants including nitrogen dioxide (NO_2_), carbon monoxide (CO), sulfur dioxide (SO_2_), ozone (O_3_), particulate matter 2.5 μm in diameter (PM_2.5_), and particulate matter 10μm in diameter (PM_10_) for the same period were obtained from the Ningbo Environment Monitoring Center. Meteorological factors included monthly average temperature (MAT, °C), monthly average highest temperature (MAHT, °C), monthly average lowest temperature (MALT, °C), monthly average relative humidity (MAH, %), monthly average atmospheric pressure (MAP, hPa), and monthly average wind speed (MAS, m/s) and data were obtained from the Ningbo Meteorological Bureau.

### 2.2. Construction of the ARIMA Model

Following Li et al. [9], we constructed a seasonal ARIMA model, which was expressed as ARIMA (*p*, *d*, *q*) (*P*, *D*, *Q*)_s_. The variable *p* is the order of the autoregression (AR) process, *q* is the number of moving average (MA) terms, *d* represents the differencing process to form a stationary times series, and *P*, *D*, and *Q* are the seasonal orders of the AR, differencing, and MA processes, respectively [3]. Additionally, s denotes the seasonal period. The number of PTB incidence predicted at time *t* (*Y_t_*) was determined by the formula: $Y_{t}=\frac{\theta_{q}\left(B\right)\Theta_{Q}\left(B^{S}\right)a_{t}}{\Phi_{P}\left(B^{S}\right)\varphi_{p}\left(B\right){\left(1-B\right)}^{d}{\left(1{-B}^{S}\right)}^{D}}$, where *θ_q_*(*B*) is the operator of the moving-average model, *Θ_Q_*(*B^S^*) is the operator of the seasonal-moving average model, *φ_p_*(*B*) is the operator of the auto-regressive model, *Φ_P_*(*B^S^*) is the operator of the seasonal autoregressive model, (1*−B*)*^d^* is the component of the ordinary differences, (1*−B^S^*)*^D^* is the component of the seasonal differences, *a_t_* is white noise, and *Y_t_* is the predicted variable [9]. Based on the monthly incidence of PTB, we used the auto. arima ( ) function in R software (the R Core Team, Vienna, Austria) to undertake automatic ARIMA model selection and fitting according to Bayesian information criterion (BIC). BIC takes into account the number of observations and has a larger penalty compared with Akaike Information Criterion (AIC) [15]. When the number of observations is too large, it can effectively prevent the model from being too complicated and over-fitting. The model with the lowest BIC is defined as the optimal model. The Ljung–Box test was used to test whether the model residual sequence showed auto-correlation. Finally, we selected the optimal ARIMA model to predict PTB incidence in 2019 and mean absolute percentage error (MAPE) was used to model validation.

### 2.3. Cross-Correlation Analysis

Due to strong auto-correlations in the data, correlations of the time series of the monthly incidence of PTB with air pollution and meteorological factors were difficult to identify. In this study, a prewhitening process was applied to the data among the multiple exogenous regressors [3]. Prewhitening is used to avoid common trends between incidence and ambient factors [16]. We calculated the cross-correlation function (CCF) between the residual series from ARIMA model of the monthly incidence and ambient factors to identify the significant time lags. Based on the results from previous studies as well as biological and epidemiological plausibility, within a lag length of 12, we selected the lag periods with positive and significant values for the next step of the analysis [17].

### 2.4. Construction of the ARIMAX Model

The lag value of the statistically significant factors identified by the cross-correlation analysis was incorporated as exogenous regressors into the ARIMAX model constructed above. The ARIMAX model is described by equation: $Y_{t}=\frac{\theta_{q}\left(B\right)\Theta_{Q}\left(B^{S}\right)a_{t}}{\Phi_{P}\left(B^{S}\right)\varphi_{p}\left(B\right){\left(1-B\right)}^{d}{\left({1-B}^{S}\right)}^{D}}+X$, where *X* represents the exogenous regressor, which can be univariate or multivariate [9,16]. The other parameters are as described in the ARIMA model above. In this study, we first took a single lag period of risk factor in the univariate ARIMAX model. The coefficients of the model were estimated using the maximum likelihood method. Fitted MAPE was used to measure the performance of the ARIMA and the ARIMAX model in predicting PTB incidence from 2016 to 2018, and test MAPE was used to measure the performance in 2019. We determined the suitable univariate ARIMAX models according to four criteria: (a) the BIC value smaller than the optimal ARIMA model; (b) the coefficients of the regression term all significant (*p* < 0.05); (c) the fitted MAPE value smaller than the optimal ARIMA model; and (d) the test MAPE value smaller than the optimal ARIMA model. 

For the multivariate ARIMAX analysis, the incorporated factors were selected from the factors in the suitable univariate ARIMAX models. The suitable multivariate ARIMAX models were also determined by four criteria above. The optimal ARIMAX model should have the lowest BIC, fitted MAPE, and test MAPE values.

### 2.5. Statistical Software

We used the packages of “forecast”, “Stats”, and “ggplot2” of R 3.5.1 (the R Core Team, Vienna, Austria) (https://www.r-project.org/, accessed on 8 Janunary 2022) to estimate the ARIMA and ARIMAX models and present data visualization. The R code can be found in Appendix A. The significance level was set at 0.05.

## 3. Results

### 3.1. Descriptive Analysis

Descriptive statistics of the monthly incidence of PTB, the monthly average air pollutants concentration, and meteorological factors in Ningbo from 2015 to 2019 are listed in Table 1. Time series plots of these are shown in Figure 1. The annual PTB notification rates from 2015 to 2019 of Ningbo were 48.20/100,000, 45.81/100,000, 45.99/10,000, 46.92/100,000, and 44.14/100,000, respectively. The monthly incidence series plot showed a seasonal fluctuation. The peak incidence mainly occurred in March, April, and May, and the trough was more common in November, December, and February. The mean concentrations of PM_2.5_, PM_10_, SO_2_, CO, NO_2_, O_3_, were 35.79 μg/m^3^, 57.42 μg/m^3^, 10.90 μg/m^3^, 0.78 mg/m^3^, 37.47 μg/m^3^, and 95.26 μg/m^3^, respectively, compared with the World Health Organization air quality guideline levels of 5 μg/m^3^, 15 μg/m^3^, 40 μg/m^3^, 4 mg/m^3^, 10 μg/m^3^, and 60 μg/m^3^, respectively [18].

### 3.2. ARIMA Model

The optimal ARIMA model constructed by monthly incidence of PTB from 2015 to 2018 according to BIC was the ARIMA (0,0,0)(1,1,0)_12_ model with a BIC of 18.2228. The optimal ARIMA model for PM_2.5_, PM_10,_ SO_2_, CO, NO_2_, O_3_, MAT, MAHT, MALT, MAH, MAP, and MAS from 2015 to 2018 is listed in Table 2. The Ljung–Box test indicated that residual sequence from all models did not significantly depart from a white noise sequence (*p* > 0.05).

### 3.3. Cross-Correlation Analysis

#### 3.3.1. CCF between the PTB and Ambient Factors

After prewhitening, we calculated the cross-correlation function (CCF) coefficients between the PTB and each ambient factor at different lag periods. Table 3 shows that the PTB was positively correlated with the PM_2.5_ (3-month lag), PM_10_ (3-month lag), and CO (9-month lag) and negatively correlated with O_3_ (0-month lag), MAT (1, 3-month lag), MALT (1,3-month lag), MAH (7-month lag), and MAP (2, 11-month lag) (*p* < 0.05).

#### 3.3.2. CCF between the Air Pollutants and Meteorological Factors

We also calculated the CCF coefficients between prewhitened air pollutants concentration and meteorological factors residual series at lag 0 between 2015 and 2018. Table 4 shows that PM_2.5_ and PM_10_ were significant negatively correlated with MAT, MALT, and MAH; SO_2_ was significant negatively correlated with MALT and MAH; NO_2_ was significant negatively correlated with MALT, MAH, and MAS; O_3_ was significant negatively correlated with MAH, but significant positively correlated with MAP.

### 3.4. Univariate and Multivariate ARIMAX Analyses

Based on the cross-correlation analysis, we tested the univariate ARIMAX model by incorporating different lag periods of risk factors as exogenous regressors based on the ARIMA (0,0,0)(1,1,0)_12_ model from PTB time series. Because different ambient factors have different lag periods, from 1 to 11 months, in order to compare the performance of different univariate ARIMAX models and the optimal ARIMA model, it is necessary to make the sequence length of modeling consistent. Therefore, the monthly incidence of PTB and environmental factors with different lag periods from January 2016 to December 2018 were selected to construct the ARIMAX model and the modeling times were all 36 months.

As shown in Table 5, the ARIMAX (0,0,0)(1,1,0)_12_ model with the O_3_ (0-month lag) and ARIMAX (0,0,0)(1,1,0)_12_ model with the MAP (11-month lag) is the optimal univariate ARIMAX model in terms of the four criteria being considered. The BIC value, fitted and test MAPE of the two univariate ARIMAX models were all smaller than the ARIMA model, and the coefficients of the regression term of these two univariate ARIMAX models were all significant (*p* < 0.05). Then, we tested the multivariate ARIMAX model by incorporating both the O_3_ (0-month lag) and the MAP (11-month lag) as exogenous regressors based on the ARIMA (0,0,0)(1,1,0)_12_ model, and the coefficients of the regression term of the multivariate ARIMAX model were all significant (*p* < 0.05). Compared with the optimal ARIMA and two suitable univariate ARIMAX models, the multivariate ARIMAX model was the best fitting model, with the lowest BIC (8.1092), fitted MAPE (2.9097%), and test MAPE (9.2643%). Finally, the actual incidence, fitted and predicted incidences of PTB using ARIMA (0,0,0)(1,1,0)_12_ and ARIMA (0,0,0)(1,1,0)_12_ with O_3_ (0-month lag) and MAP (11-month lag) in Ningbo were shown in Figure 2.

## 4. Discussion

In this study, we developed and evaluated time series models to characterize the ambient factors of pulmonary tuberculosis transmission in Ningbo to inform control measures. To our knowledge, this is the first time series study to construct ARIMAX models to explore the role of both air pollution and meteorological factors in predicting PTB. We found that both air pollution and meteorological factors were associated with the incidence of tuberculosis in Ningbo with a lag effect. Additionally, the multivariate ARIMAX model that included both the ozone at lag 0 and the monthly mean atmospheric pressure at lag 11 had a better predicting performance than the inclusion of one variable. This modeling technique can be a useful tool for planning control interventions and could be implemented during routine tuberculosis surveillance in Ningbo.

Although preventive measures have made great progress, the prevention and treatment of tuberculosis still involves enormous challenges, such as increased drug resistance [19,20], dual infection of tuberculosis and AIDS [21], and increased migrant population [22]. Furthermore, urban air quality is an important potential factor in the contribution of tuberculosis infection. There is a growing body of evidence suggesting an association between air pollution exposure and PTB incidence. However, available evidence on the association of air pollution and PTB risk is inconsistent [23]. Our study found a significantly positive correlation of particulate matter with an aerodynamic diameter ≤2.5 μm (PM_2.5_) or ≤10 μm (PM_10_) with PTB incidence, which is consistent with other studies [24,25]. Moreover, we also found that CO was positively correlated with PTB incidence, while O_3_ was negatively correlated with PTB incidence, which is consistent with other studies [11,24]. Limited evidence from vitro studies has suggested that the survival rate of mice intravenously inoculated with *Mycobacterium tuberculosis* after intravenous injection of dissolved ozone was significantly higher than that of mice not treated with ozone [26]. The association between PTB and air pollution exposure varied across regions, which may be partially attributed to pollutant concentration or analytic methods. Our study site is a coastal city of South China, the concentrations of air pollutants in our study (PM_2.5_: 35.79 ± 14.73 μg/m^3^, SO_2_: 10.90 ± 3.97 μg/m^3^, NO_2_: 37.47 ± 12.25 μg/m^3^) were much lower than those in mega cities such as Beijing (PM_2.5_: 105.10 ± 80.90 μg/m^3,^ SO_2_: 48.60 ± 49.10 μg/m^3^, NO_2_: 64.20 ± 25.70 μg/m^3^) [27]. Evidence from the area with lower air pollution levels will be important to strengthen the basis for policy making [28].

Meteorological factors, including MAT, MALT, MAH, and MAP, had a negative effect on PTB with a lag effect in our study, which are largely consistent with the findings in previous studies [14,29]. Climate change affects tuberculosis through diverse pathways: changes in climatic factors such as temperature, humidity, and precipitation influence host response through alterations in vitamin D distribution, ultraviolet radiation, malnutrition, and other risk factors [30]. Our research also found that PM_2.5_ and PM_10_ were significant negatively correlated with MAT, MALT, and MAH, and O_3_ was significant negatively correlated with MAH, but significant positively correlated with MAP. The results of our study support previous findings that climatic factors could affect the incidence of tuberculosis by indirectly regulating urban air quality [14].

Our study used incidence of PTB from 2016 to 2018 and delayed ambient factors modeling to predict TB incidence in 2019 and compare it with the actual incidence. Cross-validating to calculate test MAPE could avoid the over-fitting of the model caused by incorporating ambient factors. For example, ARIMA (0,0,0)(1,1,0)_12_ + MAT (lag1) has the smallest fitted MAPE of 2.3205%, but its test MAPE is as high as 12.0531%, which is an obvious over-fitting phenomenon. The fitted and test MAPE of ARIMA (0,0,0)(1,1,0)_12_ + O_3_ (lag0) + MAP (lag11) are both smaller than the optimal ARIMA model, so over-fitting is effectively avoided. Through ARIMA model prewhitening, we found a lag correlation between the residual sequence of ambient factors and the residual sequence of tuberculosis incidence, which suggests that it is possible to control tuberculosis and other infectious diseases by protecting the environment and reducing pollutants. 

Admittedly, there were limitations in our study. Firstly, because there was a lag relationship between ambient factors and the incidence of PTB, we could accurately predict the incidence in these future lag periods by using the ARIMAX model. For example, the lag time of MAP is 11 months, so we can accurately predict the incidence of the next 11 months. However, there was no lag relationship between O_3_ and PTB in this study. Therefore, using O_3_ to predict tuberculosis in the same month does not produce significant result. In addition, the ARIMA model can only extract the seasonal and long-term trends of ambient factors and tuberculosis incidence. This model cannot extract and predict the information of residual sequences. Thus, it is invalid to use the ARIMA model to predict future ambient factors and then incorporate them into the ARIMAX model. Because we are temporarily unable to accurately predict the future ambient factors, we can only rely on factors such as MAP to accurately predict the incidence of PTB in the next 11 months. It is our future direction to improve the accuracy of the ARIMAX model in predicting long-term incidence of PTB by finding a more advanced model to predict the future ambient factors. Secondly, the ARIMAX model can only identify a correlation between PTB incidence and ambient factors but cannot identify causal relationships. Thirdly, data from fixed monitoring stations assume that all participants in the target area were at the same level of air pollution exposure, and personal exposure data were not collected [31]. 

## 5. Conclusions

Ambient air pollutants and meteorological factors were associated with monthly incidence of PTB with a lag effect. Meteorological factors may affect the incidence of PTB by indirectly regulating urban air quality. The multivariate ARIMAX model that included both the ozone with a 0-month lag and the monthly average atmospheric pressure with an 11-month lag had the best performance to predict the incidence of PTB. However, ARIMAX has limited improvement in prediction accuracy compared with the ARIMA model. This modeling technique can be a useful tool for planning control interventions and could be implemented during routine tuberculosis surveillance in Ningbo. When the peak of tuberculosis and low atmosphere pressure arise, health education on tuberculosis should be strengthened to remind people to seek medical treatment as soon as symptoms appear.

## Figures and Tables

**Figure 1 ijerph-19-05385-f001:**
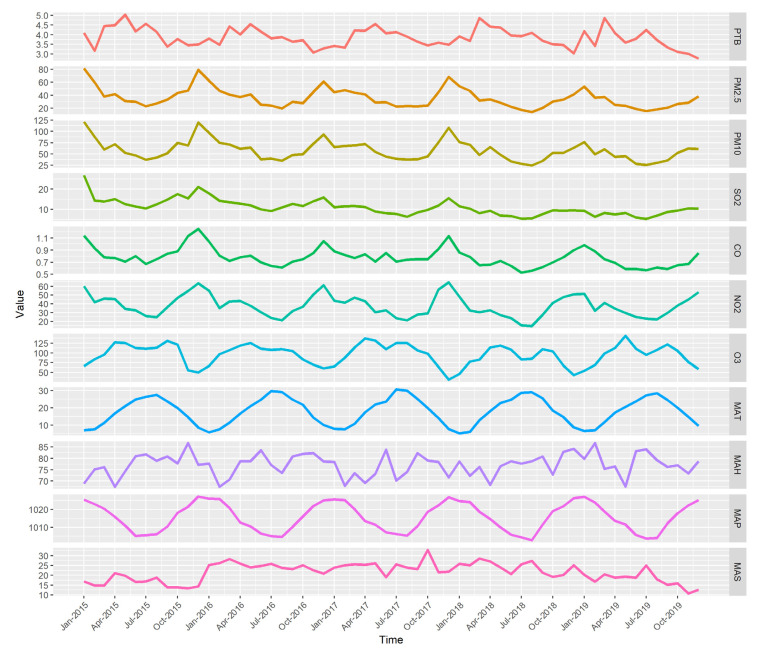
Time series of the monthly incidence of PTB, the monthly average air pollutants concentration, and meteorological factors in Ningbo from 2015 to 2019.

**Figure 2 ijerph-19-05385-f002:**
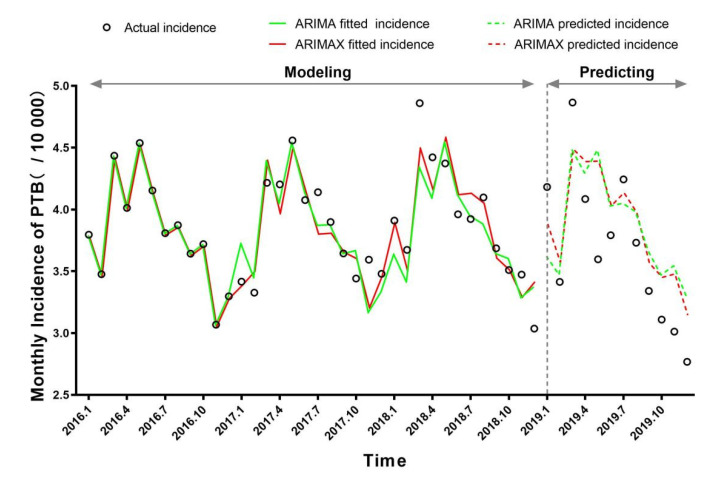
Actual incidence, fitted and predicted incidences of ARIMA (0,0,0) (1,1,0)_12_ and ARIMA (0,0,0) (1,1,0)_12_ with O_3_ (0-month lag) and MAP (11-month lag) in Ningbo.

**Table 1 ijerph-19-05385-t001:** Descriptive statistics of the monthly incidence of PTB, the monthly average air pollutants concentration, and meteorological factors in Ningbo from 2015 to 2019.

Variables	*Mean*	*S.D.*	*Min*	*P* _25_	*P* _50_	*P* _75_	*MAX*	*IQR*
Monthly incidence of PTB	3.85	0.50	2.77	3.48	3.80	4.17	5.03	0.69
Air pollutants concentration								
PM_2.5_ (μg/m^3^)	35.79	14.73	14.35	24.30	32.64	43.95	81.42	19.65
PM_10_ (μg/m^3^)	57.42	21.51	24.48	41.62	52.55	69.57	121.16	27.95
SO_2_ (μg/m^3^)	10.90	3.97	5.130	8.182	10.36	12.665	26.900	4.483
CO (mg/m^3^)	0.78	0.15	0.53	0.67	0.75	0.85	1.25	0.18
NO_2_ (μg/m^3^)	37.47	12.25	14.68	28.76	35.36	46.19	64.52	17.43
O_3_ (μg/m^3^)	95.26	26.79	31.68	71.86	104.70	113.32	143.55	41.46
Meteorological factors								
MAT (°C)	17.79	7.85	5.32	10.62	18.21	24.66	30.57	14.04
MAHT (°C)	21.88	7.97	8.61	14.66	23.25	28.24	35.97	13.58
MALT (°C)	14.53	8.01	2.83	6.35	14.20	22.06	26.74	15.70
MAH (%)	76.70	5.23	63.40	73.49	77.74	80.78	86.70	7.29
MAP (hPa)	1015.58	7.77	1002.69	1009.205	1015.89	1022.41	1027.19	15.18
MAS (m/s)	21.36	4.72	10.67	18.64	21.75	25.18	32.81	6.54

Notes: *IQR* = *P*_75_ − *P*_25_; MAT: monthly average temperature; MAHT: monthly average highest temperature; MALT: monthly average lowest temperature; MAH: monthly average relative humidity; MAP: monthly average atmospheric pressure; MAS: monthly average wind speed.

**Table 2 ijerph-19-05385-t002:** The optimal ARIMA model for monthly incidence of PTB, the monthly average air pollutants concentration, and meteorological factors in Ningbo from 2015 to 2018.

Variables	Model	BIC	Ljung–Box Test	
			*X*-Squared	*p*-Value
**Monthly incidence of PTB**	**ARIMA(0,0,0)(1,1,0)_12_**	18.2228	2.1833	0.1395
Air pollutants concentration				
PM_2.5_ (μg/m^3^)	ARIMA (1,0,0)(0,1,0)_12_	266.6336	0.7367	0.3907
PM_10_ (μg/m^3^)	ARIMA (1,0,0)(0,1,0)_12_	298.6734	0.6880	0.4068
SO_2_ (μg/m^3^)	ARIMA (1,0,0)(0,1,0)_12_	161.6301	0.5341	0.4649
CO (mg/m^3^)	ARIMA (1,0,0)(1,1,0)_12_	−74.6422	1.3417	0.2467
NO_2_ (μg/m^3^)	ARIMA (1,0,0)(0,1,1)_12_	238.207	0.7107	0.3992
O_3_ (μg/m^3^)	ARIMA (0,0,1)(1,1,0)_12_	297.4298	0.0011	0.9736
Meteorological factors				
MAT (°C)	ARIMA (0,0,0)(0,1,1)_12_	126.2939	2.2553	0.1332
MAHT (°C)	ARIMA (1,0,0)(0,1,1)_12_	148.7069	0.1210	0.7280
MALT (°C)	ARIMA (0,0,0)(0,1,0)_12_	139.3857	2.5996	0.1069
MAH (%)	ARIMA (0,0,0)(1,1,0)_12_	224.0759	0.0009	0.9757
MAP (hPa)	ARIMA (0,0,0)(1,1,0)_12_	250.4339	0.2565	0.6125
MAS (m/s)	ARIMA (0,1,1)	262.4965	0.4851	0.4861

**Table 3 ijerph-19-05385-t003:** CCF coefficients between the prewhitened ambient factors residuals series with different time lags and the prewhitened PTB incidence residuals series from 2015 to 2018.

Factors	Lag Periods (Months)												
	0	1	2	3	4	5	6	7	8	9	10	11	12
PM_2.5_	0.040	0.145	−0.125	0.353 *	0.028	0.124	−0.018	0.164	−0.175	0.151	0.013	−0.122	0.112
PM_10_	0.068	0.166	−0.114	0.343 *	0.041	0.076	−0.049	0.209	−0.172	0.190	−0.007	−0.110	0.077
SO_2_	0.202	0.069	−0.127	0.176	0.125	−0.161	−0.034	0.151	−0.211	0.031	0.103	−0.270	0.145
CO	−0.013	0.075	−0.028	−0.086	−0.111	0.100	−0.036	0.115	−0.072	0.291 *	−0.083	0.165	0.000
NO_2_	0.034	−0.100	−0.106	0.044	0.073	0.013	−0.164	0.008	−0.209	0.278	−0.075	−0.064	0.073
O_3_	−0.321 *	−0.135	−0.279	−0.085	−0.014	0.026	−0.064	0.072	0.111	0.081	0.113	−0.151	0.216
MAT	−0.206	−0.406 *	−0.102	−0.316 *	−0.002	−0.081	0.034	0.093	0.022	−0.126	−0.095	0.146	0.124
MAHT	0.093	−0.215	−0.040	−0.171	0.155	−0.014	0.148	0.241	0.162	0.080	−0.029	0.177	0.174
MALT	−0.198	−0.310 *	−0.040	−0.423 *	−0.094	−0.085	−0.077	−0.158	−0.018	−0.133	0.014	0.149	0.122
MAH	−0.136	−0.016	−0.019	−0.240	−0.238	0.073	−0.053	−0.307 *	−0.074	−0.273	−0.036	−0.085	−0.157
MAP	0.029	−0.136	−0.299 *	−0.030	0.105	−0.252	−0.151	0.066	0.053	−0.248	0.027	−0.335 *	−0.066
MAS	−0.091	0.125	−0.112	−0.205	−0.169	−0.098	0.068	0.045	0.091	−0.199	0.038	−0.053	−0.240

Note: *, *p* < 0.05.

**Table 4 ijerph-19-05385-t004:** CCF coefficients between the prewhitened air pollutants concentration and meteorological factors residuals series at lag 0 between 2015 and 2018.

Air Pollutants	Meteorological Factors					
	MAT	MAHT	MALT	MAH	MAP	MAS
PM_2.5_	−0.317 *	−0.154	−0.527 *	−0.477 *	0.022	−0.289 *
PM_10_	−0.319 *	−0.136	−0.586 *	−0.605 *	0.066	−0.248
SO_2_	−0.249	−0.059	−0.457 *	−0.529 *	0.428 *	−0.098
CO	−0.249	−0.097	−0.115	−0.059	−0.199	−0.150
NO_2_	−0.272	−0.201	−0.308 *	−0.400 *	−0.016	−0.437 *
O_3_	0.048	0.122	−0.161	−0.403 *	0.318 *	0.027

Note: *, *p* < 0.05.

**Table 5 ijerph-19-05385-t005:** Summary of the fitted parameters of the optimal ARIMA, univariate ARIMAX, and the multivariate ARIMAX model analysis in Ningbo, 2016–2018.

Model	BIC	MAPE(%)		Risk Factors				
		Fitted	Test	Vars	Coef	S.E.	T	*p*-Value
(1) ARIMA(0,0,0)(1,1,0)_12_	9.0376	3.3269	10.6693	sar1	−0.5829	0.1907	3.0566	0.0021 *
(2) ARIMA(0,0,0)(1,1,0)_12_+PM_2.5_(lag3) d	12.1682	3.3748	10.5262 d	sar1	−0.5458	0.2720	2.0064	0.0264 *
				PM_2.5_(lag3)	0.0019	0.0087	0.2147	0.4156
(3) ARIMA(0,0,0)(1,1,0)_12_+PM_10_(lag3) d	12.0662	3.3833	10.4819 d	sar1	−0.5355	0.2460	2.1767	0.0183 *
				PM_10_(lag3)	0.0018	0.0048	0.3809	0.3528
(4) ARIMA(0,0,0)(1,1,0)_12_+CO(lag9) d	11.1900	3.2306 c	9.5569 d	sar1	−0.5843	0.1903	3.0707	0.0021 *
				PM_10_(lag3)	0.0042	0.0041	1.0221	0.1570
(5) ARIMA(0,0,0)(1,1,0)_12_+O_3_(lag0) abcd	8.2634 a	3.0226 c	9.7944 d	sar1	−0.6418	0.1717	3.7374	0.0003 *
				O_3_(lag0)	−0.0061	0.0029	2.0751	0.0228 * b
(6) ARIMA(0,0,0)(1,1,0)_12_+MAT(lag1) abc	−1.4075 a	2.3205c	12.0531	sar1	−0.5673	0.1978	2.8679	0.0035 *
				MAT(lag1)	−0.1317	0.0310	4.2421	<0.0001 * b
(7) ARIMA(0,0,0)(1,1,0)_12_+MAT(lag3)	10.5244	3.3860	11.0464	sar1	−0.5146	0.2169	2.3730	0.0117 *
				MAT(lag3)	−0.0595	0.0443	1.3410	0.0944
(8) ARIMA(0,0,0)(1,1,0)_12_+MALT(lag1) abc	7.8580 a	2.8932 c	12.4087	sar1	−0.4583	0.2670	1.7165	0.0476 *
				MALT(lag1)	−0.0697	0.0357	1.9497	0.0297 * b
(9) ARIMA(0,0,0)(1,1,0)_12_+MALT(lag3) abc	8.0334 a	3.2171 c	11.4344	sar1	−0.5035	0.2217	2.2709	0.0148 *
				MALT(lag3)	−0.0736	0.0347	2.1198	0.0207 * b
(10) ARIMA(0,0,0)(1,1,0)_12_+MAH(lag7) cd	10.2365	3.1969 c	10.4618 d	sar1	−0.6337	0.1732	3.6587	0.0004 *
				MAH(lag7)	−0.0157	0.0107	1.4625	0.0764
(11) ARIMA(0,0,0)(1,1,0)_12_+MAP(lag2) c	9.9590	3.1324 c	10.9889	sar1	−0.6733	0.1607	4.1897	<0.0001
				MAP(lag2)	−0.0095	0.0059	1.6023	0.0592
(12) ARIMA(0,0,0)(1,1,0)_12_+MAP(lag11) abcd	8.4574 a	3.2063 c	10.0108 d	sar1	−0.4940	0.2186	2.2601	0.0152 *
				MAP(lag11)	−0.0125	0.0060	2.0680	0.0232 * b
(13) ARIMA(0,0,0)(1,1,0)_12_+ O_3_(lag0)+MAP(lag11) abcd	8.1092 a	2.9097 c	9.2643 d	sar1	−0.5608	0.2018	2.7791	0.0045 *
				O_3_(lag0)	−0.0054	0.0028	1.9383	0.0306 * b
				MAP(lag11)	−0.0115	0.0059	1.9461	0.0301 * b

Notes: Fitted: fitted results; Test: test results; BIC: Bayesian information criterion; MAPE: mean absolute percentage error; Coef: coefficient of risk factors; lag: time lag of risk factors; S.E.: standard error; T: t statistic; sar1: seasonal AR (1); a: meet criteria (a): the BIC value smaller than the optimal ARIMA model; b: meet criteria (b): the coefficients of the regression term all significant (*p* < 0.05); c: meet criteria (c): the fitted MAPE value smaller than the optimal ARIMA model; d: meet criteria (d): the test MAPE value smaller than the optimal ARIMA model; *: *p* < 0.05.

## Data Availability

The data used and analyzed in the current study are not publicly available because restrictions apply to the availability of the data. Data are, however, available from the corresponding author by reasonable requests, and with permission from the Ningbo Municipal CDC.

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
