# Peer review of "Modeling and Predicting Pulmonary Tuberculosis Incidence and Its Association with Air Pollution and Meteorological Factors Using an ARIMAX Model: An Ecological Study in Ningbo of China"

_ijerph, 2022, doi:10.3390/ijerph19095385_

Round 1

Reviewer 1 Report

Summary:

In this study, the authors evaluated the use of an AIRMAX model to predict TB incidence and compared model performance to AIRMA models that only use one predictor. Model performance was evaluated using the mean absolute percentage error and the BIC. The researchers found that including ozone (lag 0) and atmospheric pressure (lag 11) improved model performance.

Overall, the paper is succinct and well developed. However, the improvement between Model 1 and Model 13 seems marginal at best. This is illustrated by Figure 2 as well. I would caution against overinterpreting the results of model comparisons. The “optimal” ARIMAX model may outperform the ARIMA model, but for practical purposes the added benefits seem small. Given the noted limitations, the conclusions should be softened to reflect the modest gains of this technique.

Specific Comments:

Introduction

Line 53: It would be helpful for the authors to provide additional justification for why these environmental variables (air pollutants and meterological factors) should be included in models of TB incidence. Why would these factors be important for TB incidence? How much of TB incidence do we anticipate they will explain?

Line 97: Why what the BIC criterion used here? Please provide a justification.

Line 144: Double check the units for the WHO guideline for CO. I believe the units for CO are given in mg/m3.

Line 156: This information would be better presented in a table format.

Reviewer 2 Report

This study describes the use of ARIMAX modeling to predict PTB incidence based on meterological and air pollution factors. While potentially interesting as a method, it is unclear what is the implication of this data from the paper as it stands for the IJERPH audience. A few items to be addressed are marked on the pdf attached.
